# Kineret: Israel's Largest Hospital Network Transformed into the OMOP common data model for collaborative research

Nadav Rappoport[1,2,3*], Guy Livne[2], Naama Perry Cohen[2], Nir Makover[2], Hadas Eshel-Geva[2], Hadar Kapach[2], Tomer Hadad[2], Yarin Alon[2], Robyn Rubin[2], Segev Chai[2], Shirell da Villa[2], Ohad Hochman[4]

**1** Department of Software and Information Systems Engineering, Ben-Gurion University of the Negev, Beer Sheva, Israel, **2** The Directorate of Government Medical Centers, State of Israel Ministry of Health, Jerusalem, Israel, **3** The Data Science Research Center (DSRC), Ben-Gurion University of the Negev, Beer Sheva, Israel, **4** Bnai-Zion Medical Center, Haifa, Israel

* nadavrap@bgu.ac.il

**Data availability statement:** Aggregated data is available in the Kineret portal https://kineret.health.gov.il. Raw data is

## Abstract

**Background** In 2021, the Directorate of Government Medical Centers at the Israeli Ministry of Health launched the *Kineret* initiative to standardize clinical data across its network of public medical centers and facilitate its secondary use for research and innovation. The primary goals were to streamline data extraction, cleaning, and sharing processes, thereby enabling efficient reuse of clinical data for translational and collaborative research. The Directorate oversees a national network of 25 government healthcare institutions, including 11 general medical centers, 9 mental health centers, and 5 geriatric care facilities.

**Methods** Following an evaluation of existing data models, the Observational Medical Outcomes Partnership Common Data Model (OMOP CDM) was selected as the standard framework for semantic harmonization across institutions. A dedicated instance of ATLAS, the OHDSI open-source platform for observational research, was deployed on a secure cloud environment accessible to authorized researchers within the network. This infrastructure enables efficient feasibility assessment and exploratory data analysis buy the end users. Approved research projects are conducted within a secure, cloud-based virtual environment that supports diverse computational needs.

**Results** As of 2025, six medical centers have been successfully integrated into the *Kineret* data infrastructure, with full harmonization of their clinical data into the OMOP CDM. A seventh center is currently in the final stages of integration and is expected to join the network by the end of the year.

**Conclusion** *Kineret* initiative provides a scalable, secure, and standardized data infrastructure that supports both intra-national multi-center research and international collaborative studies. By enabling streamlined access to high-quality, harmonized clinical data, *Kineret* holds significant potential to advance both local and global healthcare research. A detailed description of the data and platform is available at https://kineret.health.gov.il/.

available upon IRB approval and a signed agreement.

**Funding:** The Israel Innovation Authority, the Israeli Ministry of Health, and the European Health Data & Evidence Network (EHDEN).

**Competing interests:** The authors have declared that no competing interests exist.

## Introduction

In recent years, the medical research landscape has undergone a significant transformation, with data sharing and collaborative learning becoming central to advancing healthcare knowledge and outcomes. The integration of hospital data into research databases is a critical step in this process, enabling researchers to harness vast amounts of clinical information for analysis and discovery [1]. However, the heterogeneity of healthcare data systems across institutions creates significant challenges for data harmonization, collaboration, and comparative studies [2]. Moreover, medical centers face substantial operational bottlenecks that delay studies. Limited technical resources and insufficient specialized personnel trained in clinical data extraction frequently resulted in extended delays, with investigators often waiting a long time to access relevant clinical information from the electronic health record systems or even abandoning studies. These constraints particularly affected cross-institutional research, though even single-site studies suffered from inefficiencies as investigators depended on overburdened informatics teams for relatively straightforward data queries [3].

To address this challenge, adopting international common data models has gained traction in the medical research community. These standardized frameworks provide a uniform structure for organizing and representing healthcare data, facilitating interoperability and simplifying collaboration across institutions worldwide. Several common clinical data models have been developed and implemented, including the Observational Medical Outcomes Partnership Common Data Model (OMOP CDM) [4], the Patient-Centered Outcomes Research Network (PCORnet) Common Data Model, the Sentinel Common Data Model [5], and the i2b2 (Informatics for Integrating Biology and the Bedside) data model [6]. Each CDM has advantages and limitations [7–10].

Among these, the OMOP CDM, adopted by the Observational Health Data Sciences and Informatics (OHDSI) collaborative, has gained widespread adoption. OHDSI, an international network of researchers and observational health databases, aims to improve health by empowering a community to collaboratively generate the evidence that promotes better health decisions and better care. The OMOP CDM standardizes the format and content of observational data, enabling large-scale analytics and facilitating the development and application of reliable evidence-based algorithms.

Healthcare data represent a vast and valuable resource for medical research, public health initiatives, and health policy development. However, these data are often stored in disparate systems with varying structures and terminologies, making it challenging to conduct comprehensive analyses or compare findings across different regions or countries. The process of converting governmental healthcare data into a standardized format, such as the OMOP CDM, holds immense significance. This transformation unlocks the potential for more robust and extensive studies and fosters international collaborations.

There is the advantage of transforming institute-wide data to OMOP CDM and not only on a per-study/per-cohort basis. As once the data is transformed once, it can be used for multiple studies, in multiple clinical fields [11–13]. Several country-wide efforts have been reported to transform and standardize EHRs to the OMOP CDM. In Estonia, a national effort was made to transform EHR data from three national health databases [14]. In the UK, a large-scale transformation of linked EHR data from multiple national sources to OMOP CDM was undertaken to convert over 1 billion rows of data from over 216 million encounters across three EHR sources [15].

In Israel, the Ministry of Health (MOH) regulates health services. In addition, MOH owns and managed the largest network of hospitals in Israel through the Directorate of Government Medical Centers established in 2015. The networks composed of 11 generic hospitals (Barzilai,

Bnei-Zion, Hille-Yafe, Wolfson, Ziv, Galil, Tzafon, Rambam, Sheba, Shamir, Ichilov), 8 Psychiatric centers (Abarbanel, Beer-Yaakov, Beer-Sheva, Jerusalem Center, Mazor, Karmel, Lev-Sharon, Shaar-Menashe), and 5 Geriatric (Dorot, Fleeman, Rishon-Lezion, Shoham, Shmuel Harofe). In 2019, the Directorate of Government Medical Centers launched an initiative to empower research and innovation using data from its hospitals. The name of the initiative is *Kineret*, which, on the one hand, is the Hebrew name for the sea of Galilee, representing the data lake. On the other hand, *Kineret* is also an abbreviation in Hebrew for medical data mining for insights (כנרת - נתונים רפואיים לתובנות כריית). For this purpose, the Directorate of Government Medical Centers developed a pipeline for the ETL (Extract Transform Load) process in collaboration with the hospitals.

The extraction, transformation, and loading (ETL) process presented several unique challenges that distinguished this implementation from typical data harmonization efforts. Unlike projects focused on specific cohorts or clinical departments, this initiative encompassed comprehensive electronic health records (EHR) representing the full spectrum of clinical services provided by Israeli hospitals, including emergency care, inpatient treatment, and ambulatory services. The heterogeneity of legacy systems—including NAMER (SAP Environment, Health, and Safety (EHS) Management - https://www.sap.com/assetdetail/2021/09/3600122d-f87d-0010-bca6-c68f7e60039b.html), Chameleon (Local Israeli vendor by Elad-Health - https://elad-health.com), Picture Archiving and Communication System (PACS), and additional platforms—significantly increased the technical complexity of the integration process. Further complicating the ETL workflow was the prevalence of Hebrew-language clinical documentation without complete mapping to international standard vocabularies, necessitating extensive cross-language terminology harmonization. To optimize computational resource allocation, the implementation strategy employed a hybrid processing architecture wherein only essential operations were performed on-premise, with the majority of data processing occurring in cloud-based environments. This architectural decision subsequently necessitated adaptation of existing regulatory frameworks to accommodate secure cloud-based storage and processing of protected health information within the constraints of Israeli healthcare governance standards.

This manuscript describes the methodological approach and technical considerations involved in transferring governmental healthcare data into the OMOP CDM v5.3. By elucidating this process, we aim to provide a blueprint for other institutions and governmental bodies seeking to enhance the utility and accessibility of their healthcare data for research purposes. Furthermore, this work contributes to the broader goal of creating a global network of standardized healthcare databases, ultimately accelerating the pace of medical discovery and improving patient care.

## Methods

The design of the ETL process and its implementation was composed of several steps. First, data profiling was executed. Then, characterization and definition of the transformation process was made by experts in the clinical notes and coders. Based on their documentation and specifications, a team of data engineers and dev-ops implemented the transformation and the load process. Tests were implemented along the process. Tests were composed of logical and quantitative tests to ensure that no records were missed or duplicated. The standard tests we used are the Data Quality Dashboard (DQD - https://github.com/OHDSI/DQD) and Achilles - https://github.com/OHDSI/Achilles) from OHDSI. On top of that, we add many more tests. For example, a quantity comparison of the source data and the target data.

## Care sites' legacy EHR systems

Each medical center in the network has its own instance of EHR system. They were all historically based on NAMER (SAP EHS Management) as the main EHR system, and in the last 4 years, gradually switched to Chameleon (Local Israeli vendor by Elad-Health - https://elad-health.com). Other EHR systems include MAX (a site-specific local development of an EHR), and dedicate Lab systems. Although the base infrastructures and EHR systems are managed on the network-level, yet local adaptation and customization are made. Therefore, the extraction part was mostly shared, but we had to make adaptations for site-specific systems.

## Extract, transform, load process

The ETL process for transferring clinical data from a hospital's on-premise legacy system to the OMOP CDM comprised several pivotal stages to guarantee both compliance with data protection regulations and the integrity of the data. Initially, the data extraction is conducted on-premise within each hospital, where sensitive patient information is de-identified in accordance with stringent local privacy standards. This step ensured that no identifiable information was exposed during subsequent processing. Each individual receives a hashed, de-identified id that is consistent across all tables and databases. This consistency is possible as patients in Israel are identified by their governmental identification number (similar to the United States' social security number). The newly introduced unique patient identifier replaces the actual identifier in each table consistently. Other identifier values, such as phone numbers and addresses, are completely removed on-premise before data is uploaded to the cloud.

Once de-identified on-premise in every site, the data is transferred to AWS dedicate environment for each site to underwent a series of transformations to align with the OMOP CDM schema and code (standard medical terms), including restructuring data, mapping local unique coding and terminology to standardized OMOP terminologies, and ensuring consistency and completeness across datasets. These transformations were performed in a secure cloud environment, which provided the necessary computational resources and security measures to handle large-scale clinical data. Finally, the transformed data is loaded into a data base in OMOP CDM format, enabling seamless integration with other datasets and facilitating advanced analytics and research within the secure cloud infrastructure.

Technically, our process is starting with extracting the data from the source systems. Then we are deidentifying the personal health information (PHI) on-premise with a local development Spark-based program. The next phase was copying the anonymized data from the on-premise to AWS S3 buckets over secured Site-To-Site VPN and AWS DataSync service. The rest of the ETL is executed on AWS. We control the workflow using Airflow that is installed on AWS as well and the Airflow execute API's calls to the on-premise processes from the extraction up to the last phase of the transformation.

## Unique ID solution

A significant challenge in multi-institutional healthcare networks, such as the Israeli Ministry of Health system, is the secure linkage of patient records across different care providers. To address this challenge, we implemented a Generated Key (GK) methodology for patient identification. The GK approach enables the creation of unique patient identifiers by combining multiple data sources from disparate legacy systems into unified OMOP CDM tables.

This mechanism was successfully employed to generate all entity identifiers within the CDM, ultimately enabling the integration of patient data from six hospitals into a single patient record ensures patient privacy, data uniqueness, and consistency, while maintaining the ability to conduct comprehensive cross-institutional analyses.

## Periodical updates

The maintenance of the OMOP CDM requires a comprehensive refresh strategy that differs from traditional data warehouse approaches. We implemented periodic full-load updates of the entire database for three critical reasons. First, the OHDSI community regularly updates the ATHENA vocabulary mappings, with newly and accuracy of code translation, maintain consistency, and up-to-date translation in historical data is essential for retrospective study. Second, source medical records frequently undergo retrospective modifications, with changes occurring days or even months after initial documentation. These retrospective modifications can have cascading effects beyond the immediately affected records, thus a delta update mechanism that is the common data warehouse approach can lead to wrong and partial patient medical history information. and Third, for enhanced privacy, a new set of GKs is generated in all tables with each load. To address these challenges, we developed an automated ETL pipeline that performs complete data refresh cycles on a periodic basis, including comprehensive quality assurance testing. This approach ensures that any modifications to the ETL process, such as the mapping of newly encountered source codes, are uniformly applied across the entire historical dataset rather than being limited to recent records. This methodology maintains data consistency and standardization across the temporal spectrum of our clinical data.

## Terminologies mapping

The standardization of medical terminologies across multiple healthcare centers presented several significant challenges in our OMOP CDM implementation: First, the Israeli Ministry of Health has established a local catalog called IC, which extends the ICD9-CM diagnosis coding system. This proprietary extension is not included in the ATHENA standardized vocabularies and cannot be processed through USAGI, the OHDSI terminology mapping tool [16]. Second, individual hospitals maintain the authority to modify and expand their local vocabularies. Third, a portion of the codes and their descriptions are documented in Hebrew, adding a layer of complexity to the standardization process. To address these challenges, we implemented a two-phase mapping approach. In the initial phase, we developed a language-based model to establish correspondences between each medical center's vocabulary codes and descriptions with OMOP CDM standard concepts. This automated approach achieved a mapping accuracy exceeding 0.9 for approximately 85% of the vocabulary terms. For the remaining 15% of terms, we employed a manual mapping process utilizing domain experts and the USAGI tool, developed by the OHDSI collaborative [16]. USAGI facilitates the conversion of data warehouse terminology to standardized OMOP concepts, but requires expert oversight to ensure accurate mapping. We strategically engaged domain specialists: laboratory professionals for laboratory terms, pharmacists for medication-related terminology, and physicians for diagnostic an procedures codes. Each specialist independently conducted mappings within their respective domains of expertise. Through this combined approach, we successfully mapped over 64,000 source concepts to approximately 33,000 standard target concepts (Table 1). The condition and procedure catalogs comprised the largest proportion of

**Table 1. Summary of source vocabularies and number of unique source codes in each and the number of unique mapped target codes.**

|  | Source vocabulary ID | #codes in source | #mapped target codes |
|---|---|---|---|
| 1 | ANTIBIOTIC_SENSITIVITY | 4 | 4 |
| 2 | AN_MOH | 343 | 77 |
| 3 | CHAMELEON_ALLERGIES | 56 | 27 |
| 4 | CHAMELEON_CONDITION_STATUS | 10 | 2 |
| 5 | CHAMELEON_DRUGS | 3661 | 2766 |
| 6 | CHAMELEON_DRUG_SENSITIVITY | 328 | 141 |
| 7 | CHAMELEON_HABIES | 5 | 5 |
| 8 | CHAMELEON_MED_ROUTE | 87 | 36 |
| 9 | CHAMELEON_SCORE | 51 | 45 |
| 10 | CHAMELEON_SURG_ROLE | 7 | 5 |
| 11 | CHAMELEON_TIME_STAMP | 3 | 3 |
| 12 | CHAMELEON_TPN | 615 | 632 |
| 13 | CHAMELEON_VITAL_SIGNS | 315 | 265 |
| 14 | COUNTRY_OF_ORIGIN | 237 | 227 |
| 15 | MARTIAL_STATUS | 5 | 5 |
| 16 | MAX_VITAL_SIGNS | 4 | 4 |
| 17 | MICRO_CULTURE_RESULTS | 2 | 14 |
| 18 | NAMER | 9 | 9 |
| 19 | NAMER_ADMITTED_FROM | 61 | 19 |
| 20 | NAMER_ANTIBIOTIC_SUSCEPTIBILITY | 151 | 137 |
| 21 | NAMER_CATALOG_1 | 10666 | 1392 |
| 22 | NAMER_CONDITION | 17207 | 10361 |
| 23 | NAMER_CONDITION_STATUS | 5 | 5 |
| 24 | NAMER_DEPARTMENTS | 1852 | 164 |
| 25 | NAMER_DISCHARGE_TO | 17 | 5 |
| 26 | NAMER_DISCHARGE_TO_MOVEMENT | 6 | 1 |
| 27 | NAMER_DISCHARGE_TO_RELEASE | 16 | 10 |
| 28 | NAMER_DRUGS | 6734 | 4410 |
| 29 | NAMER_GENDER | 2 | 2 |
| 30 | NAMER_HY | 52 | 44 |
| 31 | NAMER_LABS | 6633 | 3307 |
| 32 | NAMER_LABS_VALUE | 2 | 2 |
| 33 | NAMER_LAB_SPECIMEN | 360 | 60 |
| 34 | NAMER_LOCATIONS | 1139 | 261 |
| 35 | NAMER_MED_ROUTE | 81 | 45 |
| 36 | NAMER_MODIFIERS | 14 | 3 |
| 37 | NAMER_OPERATOR | 5 | 5 |
| 38 | NAMER_PROCEDURES | 11725 | 6746 |
| 39 | NAMER_RISK_FACTORS | 363 | 326 |
| 40 | NAMER_TASKS | 208 | 41 |
| 41 | NAMER_TIMESTAMPS | 29 | 16 |
| 42 | NAMER_TPN | 321 | 277 |
| 43 | NAMER_VISIT_DETAIL_ADMITTING | 15 | 3 |
| 44 | NAMER_VISIT_DETAIL_DISCHARGE | 8 | 6 |
| 45 | NAMER_VISIT_DETAIL_SOURCE | 7 | 5 |
| 46 | NAMER_VITAL_SIGNS | 63 | 36 |
| 47 | NAMER_ZBACTERIUM | 683 | 631 |
| 48 | NURSE_SPECIALITY | 23 | 13 |
| 49 | OTHER_SPECIALITIES | 251 | 75 |
| 50 | PHYSICIAN_SPECIALITIES | 60 | 57 |
| 51 | RELIGION | 7 | 5 |
| 52 | ROLE_MOH | 23 | 14 |
| 53 | UNIT_CONCEPT | 232 | 109 |

source concepts. While the medical centers largely shared common source codes, each facility maintained unique extensions to address local requirements (Table 2). The mapping is an ongoing task with new codes in each periodical load.

**Diagnoses.** The source vocabulary is based on ICD-9-CM, in addition to local modifications and additions. Therefore, the source ICD-9-CM concepts were mapped using the mapping tables available in Athena [17]. The rest of the terms were mapped by experts using USAGI [16].

**Procedures.** The source vocabularies of the procedures are CPT-4 and ICD-9CM. Therefore, the mapping from source concepts to target standards concepts was performed using the mapping tables from Athena [17].

**Lab results.** The source vocabulary of the laboratory test names is LOINC and, therefore, was mapped using the mapping table from Athena [17]. The modifiers and units concepts were manually mapped to standard concepts using Athena.

**Medications.** The source vocabulary of medications is a local catalog. Moreover, many drugs have a commercial Israeli brand name which is not part of any international vocabulary. Therefore, most drugs' mapping was conducted using our automated approach. The non-mapped drugs were manually mapped to the standard vocabulary RxNorm [18] by pharmacists according to active ingredients, ATC concepts, and routes of administration. Routes of administration were manually mapped to standard concepts using USAGI to fill in the *route_concept_id* field.

**Other concepts.** Other concepts that are originally in Hebrew like modifiers (like bilateral procedure) and types (like *visit_type_concept_id* and *condition_type_concept_id*) were manually mapped by querying Athena.

## Non-OMOP data

One limitation of OMOP CDM is that it is a structured database which does not cover all data available in the EHR. For example, images, ultrasound recordings, PDF files, ECG, and fetal monitor signals. In order to enrich *Kineret* datalake we created processes that extract such types of data from the source systems. The non-OMOP items are automatically assigned to the appropriate de-identified visits and patients, so the data is enriched appropriately. This data is not periodically updated, but rather extracted upon request and approval of studies. Therefore, the data is not stored in a central placed, but upload directly to the virtual research room, and deleted when the study is over.

## Provision of a virtual research machine

The OMOP data of each medical center is uploaded to an individual PostgreSQL instance on Amazon Web Services. One ATLAS [19] instance on the cloud is connected to the individual

Table 2. **Summary of the number of unique source codes and mapped target codes by medical center.**

| Medical center | #unique source codes | #unique target codes | #unmapped source codes |
|---|---|---|---|
| Barzilai | 54766 | 27949 | 14 |
| Bnai-Zion | 55222 | 28202 | 13 |
| Hillel Yaffe | 55680 | 28037 | 13 |
| Nahariya | 54901 | 27994 | 13 |
| Tzfon | 56398 | 28318 | 13 |
| Shamir | 56097 | 28351 | 13 |

databases, this allows each and every researcher to validate data viability for the study in all medical centers at once in a form that one script fits all (Fig 1).

Once a data application is approved by the IRB committee and the PIs had signed the agreement with *Kineret*, they get a virtual machine for analysis and development with the approved data from one or more medical centers.

## Results

ETL process was developed and executed in multiple rounds for enhancing the correctness and completeness of the process as well as improving the quality of the data. We provide a diagram illustrating the overall architecture of the system (Fig 2). A summary of the row numbers in the OMOP database is given in Table 3.

In total, in the mapped data there were 3,287,140 female patients, 3,539,384 male patients, and 10,163 others (Table 4). The distribution of year of birth of patients in *Kineret* is provided in Fig 3.

### Visit types

The medical centers in the network provide inpatient and outpatient care. Most visits are out-patient visits (about 29M, 55%), then emergency room visits (15M visits, 28%), and the rest are inpatient visits (8M, 16%) (Table 5).

### Death records

Death records are updated in the source from the governmental Population and Immigration Authority. In this way, death records are updated even when a patient passes away not during a visit. Most death records (78%) in our datalake are from governmental reports (Table 6).

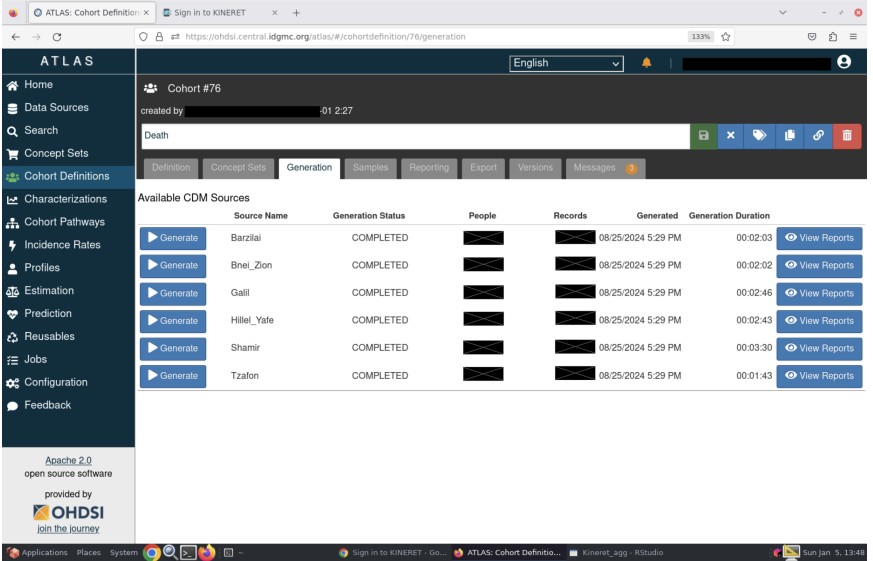

**Fig 1. Screenshot of the centralized ATLAS instance of *Kineret*.** A screenshot from the Cohort Definition tab showing the different available hospitals.

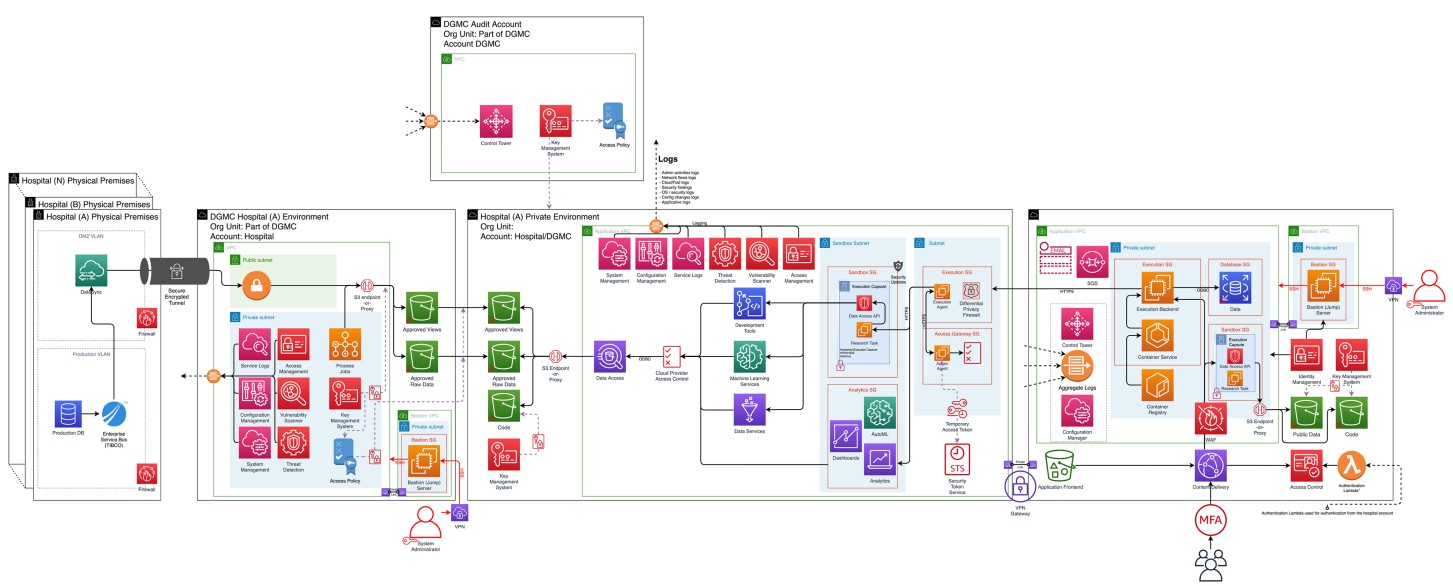

**Fig 2. A diagram describing *Kineret* system.** The process include multipel steps including on-premise (in medical centers) computing, on a centralized cloud service, and private virtual research room managed by *Kineret*.

**Table 3.** Number of records per OMOP CDM table.

| Table | #Rows |
|---|---|
| care_site | 1,263 |
| conditions | 70,123,985 |
| death | 603,271 |
| device_exposure | 3,613,875 |
| drug_exposure | 97,626,180 |
| measurements | 1,358,142,289 |
| observation | 38,794,747 |
| persons | 6,836,687 |
| procedures | 82,888,358 |
| provider | 47,541 |
| specimen | 51,436,729 |
| visits | 52,289,122 |
| visit_details | 109,964,822 |

**Table 4.** Number or persons per sex.

| Sex | #Persons |
|---|---|
| FEMALE | 3,287,140 |
| MALE | 3,539,384 |
| No matching concept | 10,163 |

## Discussion

The *Kineret* initiative represents a significant advancement in healthcare data standardization and research facilitation within Israel's medical ecosystem. By establishing a centralized data platform in the OMOP Common Data Model format, *Kineret* addresses several fundamental challenges in clinical research while creating new opportunities for collaboration

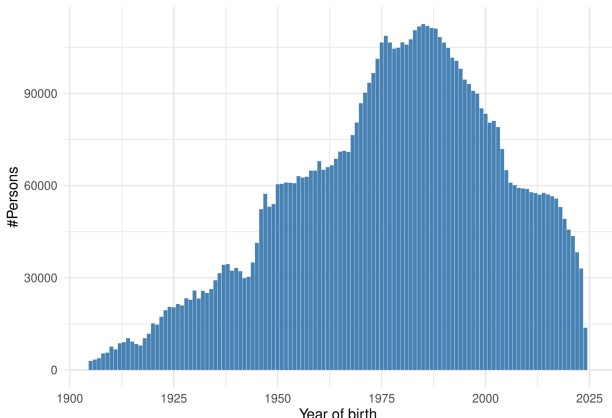

**Fig 3. Number of persons by year of birth.** Histogram of number of persons in *Kineret* born in every year from 1900 to 2025.

**Table 5**. **Number of records per visit type.**

| Visit type | #Rows |
|---|---|
| Emergency Room Visit | 14,786,492 |
| Inpatient Visit | 8,537,702 |
| No matching concept | 2 |
| Outpatient Visit | 28,964,926 |

**Table 6**. **Number of records per death type.**

| Death record type | #Persons |
|---|---|
| EHR discharge summary | 84,241 |
| Government report | 472,274 |
| No matching concept | 46,756 |

EHR - Electronic Health Records.

and discovery. By centralizing the Extract, Transform, Load (ETL) process and standardizing data through the *Kineret* initiative, one of the primary barriers to initiating research—data acquisition—has been significantly reduced. Previously, researchers were required to coordinate with IT personnel to define and execute specific database queries, often leading to delays and inefficiencies. With the implementation of *Kineret*, researchers can now independently explore and query the data using user-friendly graphical tools such as ATLAS, greatly enhancing accessibility and accelerating the research workflow.

## Research access and workflow

The process for researchers to access *Kineret* data has been streamlined to balance efficiency with appropriate oversight (Fig 4). Internal researchers can begin with feasibility testing through the ATLAS interface, allowing them to refine research questions and determine appropriate cohorts before formal study initiation. External researcher, need to contact *Kineret* stuff to perform the feasibility test. They can provide a detailed description of the cohort, or build on their own using a public ATALS instance and share the export JSON file. This preliminary step reduces the resources expended on studies that might otherwise prove infeasible due to insufficient sample sizes or data availability. Once a researcher identifies a

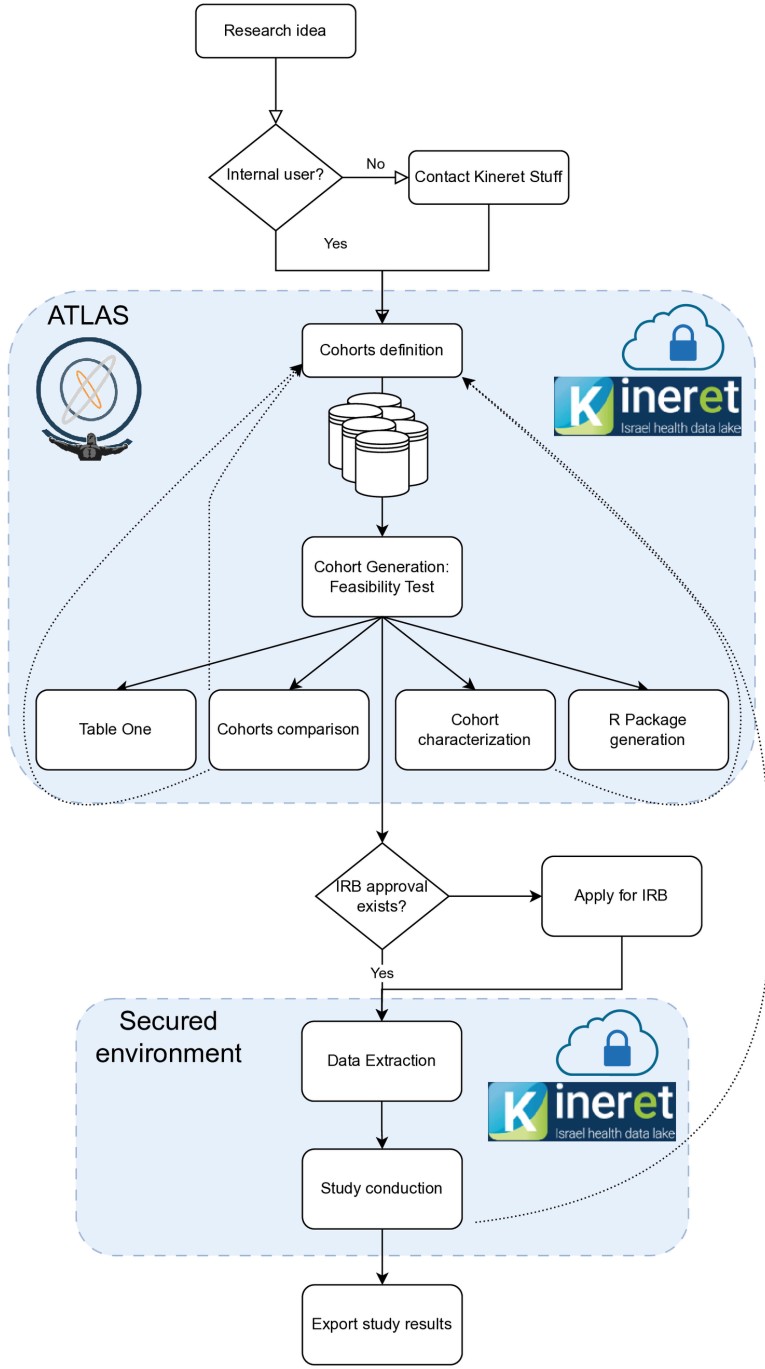

**Fig 4. Illustration of process flow in *Kineret*.** Internal users can access a secured environment with an ATLAS instance to generate cohorts from all sources at once or independently. To extract a cohort's data and upload it to the secured study room, IRB approval is required. *Kineret* stuff supports every step if required by the user.

viable study, they submit an application to the relevant Institutional Review Board (IRB). A notable regulatory advancement has transformed this previously cumbersome process. Previously, multi-center studies required separate IRB applications at each participating institution,

resulting in redundant fees, administrative burden, and significant delays. The new stream-lined approach allows researchers to submit a single primary IRB application, with other centers able to ratify this approval through an expedited internal process. This regulatory innovation significantly reduces barriers to multi-center research and promotes more collaborative investigations. Upon IRB approval and execution of the *Kineret* data use agreement, researchers receive access to a secure virtual machine with the approved dataset. This environment provides the necessary analytical tools while maintaining appropriate data governance and security controls. The cloud-based infrastructure offers scalability to accommodate varying computational requirements across different research projects, from small observational studies to complex machine learning applications.

## Technical infrastructure and tools integration

*Kineret*'s implementation has benefited substantially from integration with open-source tools developed by the OHDSI community and other stakeholders. These tools support the entire research lifecycle. USAGI facilitates concept mapping between local terminologies and standard vocabularies, addressing the particular challenges of Hebrew-language clinical documentation and proprietary extensions to standard coding systems. DQD and Achilles provide comprehensive quality assurance, enabling continuous monitoring and improvement of data quality metrics. ATLAS and PatientExploreR [20] offer intuitive interfaces for cohort definition, characterization, and exploratory analysis, allowing researchers to interact with the data without requiring advanced programming skills.

The platform's cloud-based architecture offers several advantages beyond mere technical convenience. It enables secure access from any location, facilitating collaboration between researchers at different institutions and even international partnerships. The scalable infrastructure accommodates the growing volume of clinical data and increasing computational demands for advanced analytics, without requiring substantial capital investments in local hardware.

## Impact on research and clinical practice

Since its inception, *Kineret* has supported many local and international studies that have yielded significant findings [21–23]. The standardized data structure has proven particularly valuable for cross-institutional comparisons, enabling investigations into practice variations and outcomes across different healthcare settings within the Israeli healthcare system. The initiative also addresses a critical bottleneck in clinical research: the time and technical expertise required to extract, clean, and prepare data for analysis. By providing pre-processed, standardized data, *Kineret* allows clinical researchers to focus on hypothesis generation and testing rather than data engineering. This shift has the potential to accelerate the translation of clinical insights into practice improvements. Furthermore, the OMOP CDM implementation facilitates participation in international research networks and consortia, positioning Israeli healthcare institutions within the global research community. This integration enables Israeli researchers to contribute to and benefit from large-scale, multinational observational studies that would be infeasible within a single healthcare system.

## Collaborative work

As noted, the *Kineret* initiative was built upon the formation of a broad coalition comprising eight different entities—seven independent medical centers and the directorate of government

medical centers that oversees them. Establishing this partnership required numerous meetings and discussions with hospital CEOs, financial officers, medical directors, and administrative managers. Each institution had its own concerns and resistance, necessitating a high degree of creativity to navigate the complexities.

As part of the solution, it was decided to establish a governance structure that included representatives from each hospital. Two dedicated forums were created: one focused on financial regulation, known as the Economic Regulation Committee, and another addressing clinical aspects, called the Research Forum. Each forum was staffed with appropriate representatives whose primary roles were to make joint decisions and advance the initiative both collectively and at an institutional level. This structured approach was crucial in fostering trust among all stakeholders.

At the outset, strict boundaries were set between the data lakes of each organization, ensuring full segregation—each internal researcher could access only their own hospital's data. However, after approximately a year and a half of collaboration and trust-building, all institutions approved expanding access, allowing internal researchers at one medical center to view data from across the entire network.

## Challenges and limitations

Despite its successes, the *Kineret* initiative has encountered several challenges worth noting. The standardization of terminologies across different healthcare settings required significant investment in automated mapping tools and expert review. While our approach achieved high mapping accuracy for most terms, approximately 15% required manual curation, highlighting the complexity of terminology harmonization across linguistic and institutional boundaries. One limitation of the OMOP CDM is its focus on structured data, which may not fully capture the richness of clinical information contained in unstructured notes, images, and other multimedia data. To address this limitation, we developed complementary processes to extract and link non-OMOP data types with patient records, providing a more comprehensive view of clinical information. However, these unstructured data elements present ongoing challenges for de-identification, standardization and analysis. The quarterly full-load update strategy, while ensuring data consistency, introduces computational overhead and temporary unavailability during update cycles. As the volume of data continues to grow, optimizing this process while maintaining data integrity will remain an important consideration.

## Future directions

The *Kineret* initiative continues to evolve along several trajectories. Expansion to include additional medical centers within the Israeli healthcare system remains a primary goal, with the seventh medical center expected to be fully integrated during 2025. This expansion will further enhance the representativeness and statistical power of the dataset. Future development will focus on integrating advanced analytical capabilities directly within the *Kineret* platform, including natural language processing tools for unstructured clinical notes and machine learning frameworks for predictive modeling. These enhancements will enable researchers to leverage both structured and unstructured data for comprehensive clinical investigations. We also anticipate deeper integration with international research networks through the OHDSI consortium and the European Health Data & Evidence Network (EHDEN). These collaborations will facilitate participation in large-scale, distributed research studies addressing global health challenges.

## Conclusion

In conclusion, the *Kineret* initiative represents a significant advancement in the standardization and accessibility of healthcare data within Israel's medical landscape. By adopting the OMOP CDM, *Kineret* has effectively transformed disparate clinical data from multiple medical centers into a unified format, facilitating enhanced research capabilities and collaborative learning. This initiative not only streamlines data extraction and analysis but also supports both national and international multi-site studies, thereby broadening the scope of potential research endeavors. The successful integration of data from various healthcare institutions underscores the importance of standardized frameworks in overcoming challenges related to data heterogeneity and interoperability. As *Kineret* continues to evolve, it promises to serve as a vital resource for advancing medical research, improving patient care, and fostering innovation within the healthcare sector. Furthermore, the experiences and methodologies developed through *Kineret* provide a valuable blueprint for other institutions and countries aiming to enhance their healthcare data infrastructures, thereby contributing to a global network of standardized healthcare databases that can accelerate medical discovery and improve health outcomes.

## Legal statement

The Information Protection Committee of the Israeli Directorate of Governmental Medical Centers has approved the cloud-based architecture of the Kineret Project, including the medical data anonymization process, on April 2nd, 2024. This approval was granted in accordance with legal requirements and the guidelines of the Israeli Ministry of Health.

## Acknowledgments

We would like to acknowledge all *Kineret* team at the Directorate of Government Medical Centers, Israeli Ministry of Health. Specifically, Dr. Orly Weinstein's contribution in the first steps of this initiative. The steering committee of *Kineret*: Prof. Amos Katz, Prof. Mati Berkovitch, Prof. Zehava Vadasz, Prof. Avi Peretz, Dr. Khetam Hussein, Dr. Dikla Dahan Shriki, and Prof. Aviram Nissan. We would like to thank the chairs of the institutional IRB committees: Prof. Amos Katz, Prof. Mati Berkovitch, Prof. Kamal Hussein, Dr. Noa Brar-Yannai, Dr. Efrat Wolfowitz, and Dr. Einav Yefet.

## Author contributions

**Conceptualization:** Nadav Rappoport, Guy Livne, Naama Perry Cohen, Nir Makover, Shirell da Villa.

**Data curation:** Hadar Kapach, Tomer Hadad, Yarin Alon, Robyn Rubin, Segev Chai.

**Formal analysis:** Nadav Rappoport.

**Funding acquisition:** Naama Perry Cohen.

**Methodology:** Nadav Rappoport, Guy Livne.

**Project administration:** Hadas Eshel-Geva, Hadar Kapach, Ohad Hochman.

**Resources:** Tomer Hadad.

**Software:** Segev Chai, Shirell da Villa.

**Supervision:** Nadav Rappoport, Guy Livne, Ohad Hochman.

**Writing – original draft:** Nadav Rappoport, Shirell da Villa.

**Writing – review & editing:** Nadav Rappoport, Nir Makover, Hadas Eshel-Geva, Shirell da Villa.

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
