## [Decision Letter · Decision Letter 0]

13 May 2025

PONE-D-25-16109Kineret - The Largest Israeli Chain of Hospitals' Data available now in OMOP Common Data Model Format Enabling Broad Collaborative ResearchPLOS ONE

Dear Dr. Rappoport,

Thank you for submitting your manuscript to PLOS ONE. After careful consideration, we feel that it has merit but does not fully meet PLOS ONE’s publication criteria as it currently stands. Therefore, we invite you to submit a revised version of the manuscript that addresses the points raised during the review process.

We look forward to receiving your revised manuscript.

Kind regards,

Ágnes Vathy-Fogarassy, Ph.D.

Academic Editor

PLOS ONE

 [The Israel Innovation Authority, the Israeli Ministry of Health, and the European Health Data & Evidence Network (EHDEN)]. 

3. Please update your submission to use the PLOS LaTeX template. The template and more information on our requirements for LaTeX submissions can be found at http://journals.plos.org/plosone/s/latex"

5. Please include a separate caption for each figure in your manuscript

Additional Editor Comments (if provided):

Reviewers' comments:

Reviewer's Responses to Questions

**Comments to the Author**

1. Is the manuscript technically sound, and do the data support the conclusions?

Reviewer #1: Yes

Reviewer #2: Yes

2. Has the statistical analysis been performed appropriately and rigorously? 

Reviewer #1: Yes

Reviewer #2: N/A

3. Have the authors made all data underlying the findings in their manuscript fully available?

Reviewer #1: Yes

Reviewer #2: Yes

4. Is the manuscript presented in an intelligible fashion and written in standard English?

Reviewer #1: Yes

Reviewer #2: Yes

5. Review Comments to the Author

Reviewer #1: A fascinating and both clinically and digitally relevant paper demonstrating how the largest Israeli Chain of Hosptals' Data, now available in OMOP Common Data Model Format is enabling broad collaborative research

Reviewer #2: OVERVIEW AND GENERAL RECOMMENDATION

Thank you for your submission on this important and timely topic.

The manuscript describes the implementation of the OMOP CDM as a research data repository by various healthcare providers and institutions across Israel as part of the Kinert project. A particularly innovative aspect is the nationwide adoption of OMOP CDM, including the transformation of governmental healthcare data and the use of a hybrid processing architecture. This represents a significant and promising development for the scientific community working with real-world data.

While the manuscript is generally well-written and easy to follow, the authors may wish to reconsider whether the focus of the paper should be more strongly placed on the described implementation process itself. If so, this should be more clearly reflected in the Results section. Further suggestions can be found in the “Major Comments” and “Minor Comments” sections.

Overall, this is a highly relevant and valuable contribution to the field of real-world data research. I therefore recommend a major revision of the manuscript.

MAJOR COMMENTS

1. Section 2.2: Could you please provide more detail on the de-identification process and how hashes are managed? Is there a trusted third party responsible for protecting these data?

2. Section 2.2: The execution of ETL processes in the cloud is particularly interesting for the research community. Could you elaborate further on the required computational resources and the security measures?

3. Section 2.4: Why is a full reload performed due to updates in the ATHENA vocabularies? Typically, data should be mapped to the standard concept that was valid and appropriate at the time the data was recorded, as indicated by concept validity periods. Retrospectively updating mappings to newer concepts may distort the historical accuracy of the data representation.

4. Section 2.6: Where are the non-OMOP data stored? How are these datasets linked or integrated with the OMOP CDM data?

5. Section 3: The Results section is quite short compared to the other sections. If the main focus of the paper is the implementation approach of the OMOP CDM, this should be described in the Results section rather than in the Methods. In that case, the Methods section should explain how the authors developed the presented implementation strategy.

6. Section 3: Including a diagram of the overall system architecture would be highly valuable and would significantly enhance the manuscript.

MINOR COMMENTS

7. Section 1: The limitations mentioned in the Introduction (“limited technical resources and insufficient specialized personnel”) are not directly addressed by the adoption of research data repositories like the OMOP CDM. Using OMOP CDM, and especially tools like ATLAS, also requires trained personnel. It would be helpful if the authors could briefly address this point.

8. Section 2.1: Could you please add links to additional information about the EHR systems mentioned? This would be useful for readers who are not familiar with these products and would like to learn more.

6. PLOS authors have the option to publish the peer review history of their article (what does this mean?). If published, this will include your full peer review and any attached files.

Reviewer #1: No

Reviewer #2: No

---

## [Author Response · Author response to Decision Letter 1]

18 Aug 2025

Rebuttal for the manuscript PONE-D-25-16109 ”Kineret: Israel’s

Largest Hospital Network Transformed into the OMOP Common Data Model for Collaborative Research”

Nadav Rappoport et. al

July 25th, 2025

The authors would like to thank the editor and reviewers for their constructive comments and suggestions titled Kineret - The Largest Israeli Chain of Hospitals’ Data available now in OMOP Common Data Model Format Enabling Broad Collaborative Research that have helped improve the quality of this manuscript. The resubmitted manuscript has undergone a thorough revision according to the editor and reviewers’ comments. Please see our responses below. For the reviewers’ convenience, we have highlighted significant changes in the revised manuscript in blue.

Associate Editor Comments

1. Please ensure that your manuscript meets PLOS ONE’s style requirements, including those for file naming. The PLOS ONE style templates can be found at https://journals.plos.org/plosone/s/file?id=wjVg/PLOSOne_formatting_sample_main_body.pdf and https://journals.plos.org/plosone/s/file?id=ba62/PLOSOne_formatting_sample_title_authors_affiliations.pdf

Reply: The revised manuscript uses the PLOS One template to match the style requirements.

[The Israel Innovation Authority, the Israeli Ministry of Health, and the European Health Data & Evidence Network (EHDEN)].

Please state what role the funders took in the study. If the funders had no role, please state: “The funders had no role in study design, data collection and analysis, decision to publish, or preparation of the manuscript.” If this statement is not correct you must amend it as needed. Please include this amended Role of Funder statement in your cover letter; we will change the online submission form on your behalf.

Reply: The funders had no role in study design, data collection and analysis, decision to publish, or preparation of the manuscript.

3. Please update your submission to use the PLOS LATEX template. The template and more information on our requirements for LATEX submissions can be found at https://journals.plos.org/plosone/s/latex

Reply: The revised submission uses the PLOS LATEX template as provided by the journal.

Reply: The two abstracts are now identical after some modifications made.

5. Please include a separate caption for each figure in your manuscript. Reply: The revised version has a separate caption for each figure.

Reviewer A

Reviewer CommentA.1 — A fascinating and both clinically and digitally relevant paper demonstrating how the largest Israeli Chain of Hosptals’ Data, now available in OMOP Common Data Model Format is enabling broad collaborative research.

Reply: We thank the reviewer for this comment.

Reviewer B

Reviewer 2: OVERVIEW AND GENERAL RECOMMENDATION

Thank you for your submission on this important and timely topic. The manuscript describes the implementation of the OMOP CDM as a research data repository by various healthcare providers and institutions across Israel as part of the Kinert project. A particularly innovative aspect is the nationwide adoption of OMOP CDM, including the transformation of governmental healthcare data and the use of a hybrid processing architecture. This represents a significant and promising development for the scientific community working with real-world data. While the manuscript is generally well-written and easy to follow, the authors may wish to reconsider whether the focus of the paper should be more strongly placed on the described implementation process itself. If so, this should be more clearly reflected in the Results section. Further suggestions can be found in the “Major Comments” and “Minor Comments” sections. Overall, this is a highly relevant and valuable contribution to the field of real-world data research. I therefore recommend a major revision of the manuscript.

Reply: We thanks the reviewer for his comment. We do not have the intention to give the technical details of the implementation. The fine details of the ETL process are specific to the Israeli Ministry of Health facilities, and therefore we believe than there won’t be an audience for such a manuscript. We prefer to focus on the overall process, the key points and idea in designing, developing and implementing the process and the idea of a national initiative and effort for local and global multi-sites studies.

MAJOR COMMENTS

Reviewer CommentB.1 — 1. Section 2.2: Could you please provide more detail on the deidentification process and how hashes are managed? Is there a trusted third party responsible for protecting these data?

Reply: We ellaborate in section 2.2 (in the PlOS One there is no section numbering, so section “Extract, Transform, Load Process”) regarding the de-identification process which is performed onpremise - meaning independently in every institution. There was no use of third-party for this process.

Reviewer CommentB.2 — Section 2.2: The execution of ETL processes in the cloud is particularly interesting for the research community. Could you elaborate further on the required computational resources and the security measures?

Reply: We elaborated on the process as follow: Technically, our process is starting with extracting the data from the source systems. Then we are deidentifying the personal health information (PHI) on-premise with a local development Spark-based program. The next phase was copying the anonymized data from the on-premise to AWS S3 buckets over secured Site-To-Site VPN and AWS DataSync service. The rest of the ETL is executed on AWS. We control the workflow using Airflow that is installed on AWS as well and the Airflow execute API’s calls to the on-premise processes from the extraction up to the last phase of the transformation.

Reviewer CommentB.3 — Section 2.4: Why is a full reload performed due to updates in the ATHENA vocabularies? Typically, data should be mapped to the standard concept that was valid and appropriate at the time the data was recorded, as indicated by concept validity periods. Retrospectively updating mappings to newer concepts may distort the historical accuracy of the data representation.

Reply:

Thank you for your thoughtful comment. You are correct that OMOP concepts include validity periods, and that mapping according to the validity at the time of the event can help preserve historical fidelity. However, to ensure compatibility with OHDSI tools and phenotypes—many of which rely on current standard concepts—we opt to reload and remap using the latest ATHENA vocabulary updates. This approach ensures consistent and analyzable data across our network and aligns with common OHDSI practices. We recognize this as a trade-off and are considering versioned mapping metadata to improve traceability going forward. See related discussion on the OHDSI forum https://forums.ohdsi.org/t/share-your-vocab-concept-updating-process/13286.

Reviewer CommentB.4 — Section 2.6: Where are the non-OMOP data stored? How are these datasets linked or integrated with the OMOP CDM data?

Reply: We add to the manuscrip the next clarification: “The non-OMOP data is not stored in a central placed, but upload directly to the virtual research room, and deleted when the study is over.” The dataset are linked using the same person-ids, as Kineret has the mapping from patient ids to the de-identified new ids.

Reviewer CommentB.5 — 5. Section 3: The Results section is quite short compared to the other sections. If the main focus of the paper is the implementation approach of the OMOP CDM, this should be described in the Results section rather than in the Methods. In that case, the Methods section should explain how the authors developed the presented implementation strategy.

Reply: We understand the point of view of the reviewer. However, we think that the result of Kineret can be currently described shortly. We can characterize the data by providing statistics. More emphasis was put on the Methods and the discussion where we described studies workflows, and the impact on studies.

Reviewer CommentB.6 — 6. Section 3: Including a diagram of the overall system architecture would be highly valuable and would significantly enhance the manuscript.

Reply: We thank the reviewer for this suggestion which improve the manuscript and add information to potential readers. We included a detailed diagram of the system architecture. See Figure 2.

MINOR COMMENTS

Reviewer CommentB.7 — Section 1: The limitations mentioned in the Introduction (“limited technical resources and insufficient specialized personnel”) are not directly addressed by the adoption of research data repositories like the OMOP CDM. Using OMOP CDM, and especially tools like ATLAS, also requires trained personnel. It would be helpful if the authors could briefly address this point.

Reply: Correct, the mentioned limitation were not addressed by the adoption of research data repositories. But it was addressed by centralizing the ETL process for all the members of the network by Kineret. This way, there is no need for every researcher in every institution to reach out to the local IT person for data extraction. As that was not clear, we add a description in the discussion section as follow:

By centralizing the Extract, Transform, Load (ETL) process and standardizing data through the Kineret initiative, one of the primary barriers to initiating research—data acquisition—has been significantly reduced. Previously, researchers were required to coordinate with IT personnel to define and execute specific database queries, often leading to delays and inefficiencies. With the implementation of Kineret, researchers can now independently explore and query the data using user-friendly graphical tools such as ATLAS, greatly enhancing accessibility and accelerating the research workflow.

Reviewer CommentB.8 — Section 2.1: Could you please add links to additional information about the EHR systems mentioned? This would be useful for readers who are not familiar with these products and would like to learn more.

Reply: Definitely. We add the next clarifications and references: NAMER (SAP Environment, Health, and Safety (EHS) Management), Chameleon (Local Israeli vendor by Elad-Health2), and MAX (a site-specific local development of an EHR).

---

## [Editor Report · Decision Letter 1]

2 Oct 2025

Kineret: Israel’s Largest Hospital Network Transformed into the OMOP Common Data Model for Collaborative Research

PONE-D-25-16109R1

Dear Dr. Rappoport,

We’re pleased to inform you that your manuscript has been judged scientifically suitable for publication and will be formally accepted for publication once it meets all outstanding technical requirements.

Kind regards,

Michal Rosen-Zvi

Academic Editor

PLOS ONE

Additional Editor Comments (optional):

The authors have thoroughly addressed the reviewers' concerns and significantly improved the manuscript. The revised version presents a clear and comprehensive overview of the Kineret initiative, launched by the Israeli Ministry of Health to standardize clinical data across 25 government medical centers. By adopting the OMOP Common Data Model and deploying a secure, cloud-based instance of the ATLAS platform, the initiative enables streamlined data harmonization and supports both national and international collaborative research.

As of 2025, six centers have been fully integrated into the infrastructure, with a seventh nearing completion. In response to Reviewer b’s suggestion to emphasize the implementation process, the authors clarified that their intention was not to detail the technical aspects of the ETL pipeline, which are specific to the Israeli context. Instead, they chose to focus on the broader design and development principles, highlighting the strategic vision and collaborative effort behind establishing a national infrastructure for multi-site research. This approach effectively conveys the significance of Kineret as a scalable framework for enabling both local and global studies.

The manuscript now offers a well-structured and insightful account of the infrastructure, its impact, and its potential to advance healthcare research through high-quality, harmonized clinical data.
---

## [Editor Report · Acceptance letter]

PONE-D-25-16109R1

PLOS ONE

Dear Dr. Rappoport,

I'm pleased to inform you that your manuscript has been deemed suitable for publication in PLOS ONE. Congratulations! Your manuscript is now being handed over to our production team.

Kind regards,

on behalf of

Prof. Michal Rosen-Zvi

Academic Editor

PLOS ONE